# Proto-Value Networks: Scaling Representation Learning with Auxiliary Tasks

**Jesse Farebrother** [*][1][3][5] , **Joshua Greaves** [*][5], **Rishabh Agarwal** [2][3][5], **Charline Le Lan** [4][5], **Ross Goroshin** [5], **Pablo Samuel Castro** [5], **Marc G. Bellemare** [†][3][5]

## Abstract

Auxiliary tasks improve the representations learned by deep reinforcement learning agents. Analytically, their effect is reasonably well-understood; in practice, however, their primary use remains in support of a main learning objective, rather than as a method for learning representations. This is perhaps surprising given that many auxiliary tasks are defined procedurally, and hence can be treated as an essentially infinite source of information about the environment. Based on this observation, we study the effectiveness of auxiliary tasks for learning rich representations, focusing on the setting where the number of tasks and the size of the agent's network are simultaneously increased. For this purpose, we derive a new family of auxiliary tasks based on the successor measure. These tasks are easy to implement and have appealing theoretical properties. Combined with a suitable off-policy learning rule, the result is a representation learning algorithm that can be understood as extending Mahadevan & Maggioni (2007)'s proto-value functions to deep reinforcement learning – accordingly, we call the resulting object *proto-value networks*. Through a series of experiments on the Arcade Learning Environment, we demonstrate that proto-value networks produce rich features that may be used to obtain performance comparable to established algorithms, using only linear approximation and a small number (~4M) of interactions with the environment's reward function.

## 1 Introduction

In deep reinforcement learning (RL), an agent maps observations to a policy or return prediction by means of a neural network. The role of this network is to transform observations into a series of successively refined features, which are linearly combined by the final layer into the desired prediction. A common perspective treats this transformation and the intermediate features it produces as the agent's *representation* of its current state. Under this lens, the learning agent performs two tasks simultaneously: representation learning, the discovery of useful state features; and credit assignment, the mapping from these features to accurate predictions.

Although end-to-end RL has been shown to obtain good performance in a wide variety of problems (Mnih et al., 2015; Levine et al., 2016; Bellemare et al., 2020), modern RL methods typically incorporate additional machinery that incentivizes the learning of good state representations: for example, predicting immediate rewards (Jaderberg et al., 2017), future states (Schwarzer et al., 2021a), or observations (Gelada et al., 2019); encoding a similarity metric (Castro, 2020; Agarwal et al., 2021a; Zhang et al., 2021); and data augmentation (Laskin et al., 2020). In fact, it is often possible, and desirable, to first learn a sufficiently rich representation with which credit assignment can then be efficiently performed; in that sense, representation learning has been a core aspect of RL from

---

Correspondence to: Jesse Farebrother <jfarebro@cs.mcgill.ca>

[1] McGill University, [2] Université de Montréal, [3] Mila – Québec AI Institute, [4] University of Oxford

[5] Google Research – Brain Team, [*] Equal contribution, [†] CIFAR Fellow.

Offline Reinforcement Learning Workshop at Neural Information Processing Systems, 2022

its early days (Sutton & Whitehead, 1993; Sutton, 1996; Ratitch & Precup, 2004; Mahadevan & Maggioni, 2007; Diuk et al., 2008; Konidaris et al., 2011; Sutton et al., 2011).

An effective method for learning state representations is to have the network predict a collection of auxiliary tasks associated with each state (Caruana, 1997; Jaderberg et al., 2017; Chung et al., 2019). In an idealized setting, auxiliary tasks can be shown to induce a set of features that correspond to the principal components of what is called the auxiliary task matrix (Bellemare et al., 2019; Lyle et al., 2021). This makes it possible to analyze the theoretical approximation error (Petrik, 2007; Parr et al., 2008), generalization (Le Lan et al., 2022), and stability (Ghosh & Bellemare, 2020) of the learned representation. Perhaps surprisingly, there is comparatively little that is known about their empirical behaviour on larger-scale environments. In particular, the scaling properties of representation learning from auxiliary tasks – i.e., the effect of using more tasks, or increasing network capacity – remain poorly understood. This paper aims to fill this knowledge gap.

Our approach is to construct a family of auxiliary rewards that can be sampled and subsequently. Specifically, we implement the successor measure (Blier et al., 2021; Touati & Ollivier, 2021), which extends the successor representation (Dayan, 1993) by replacing state-equality with set-inclusion. In our case, these sets are defined implicitly by a family of binary functions over states. We conduct most of our studies on binary functions derived from randomly-initialized networks, whose effectiveness as random cumulants has already been demonstrated (Dabney et al., 2021).

Although our results may hold for other types of auxiliary rewards, our method has a number of benefits: it can be trivially scaled by sampling more random networks to serve as auxiliary tasks, it directly relates to the binary reward functions common of deep RL benchmarks, and can to some extent be theoretically understood. The actual auxiliary tasks consist in predicting the expected return of the random policy for their corresponding auxiliary rewards; in the tabular setting, this corresponds to proto-value functions (Mahadevan & Maggioni, 2007; Stachenfeld et al., 2014; Machado et al., 2018). Consequently, we call our method *proto-value networks* (PVN).

We study the effectiveness of this method on the Arcade Learning Environment (ALE) (Bellemare et al., 2013). Overall, we find that PVN produces state features that are rich enough to support linear value approximations that are comparable to those of DQN (Mnih et al., 2015) on a number of games, while only requiring a fraction of interactions with the environment reward function. We explore the features learned by PVN and show that they capture the temporal structure of the environment, which we hypothesize contributes to their utility when used with linear function approximation.

In an ablation study, we find that increasing the value network's capacity improves the performance of our linear agents substantially, and that larger networks can accommodate more tasks. Perhaps surprisingly, we also find that our method performs best with what might seem like small number of auxiliary tasks: the smallest networks we study produce their best representations from 10 or fewer tasks, and the largest, from 50 to 100 tasks. In a sense, this finding corroborates the result of Lyle et al. (2021, Fig. 5), where optimal performance (on a small set of Atari 2600 games and with the standard DQN network) was obtained with a single auxiliary task. From this finding we hypothesize that individual tasks may produce much richer representations than expected, and the effect of any particular task on fixed-size networks (rather than the idealized, infinite-capacity setting studied in the literature) remains incompletely understood.

## 2 Related work

Deep RL algorithms have employed auxiliary prediction tasks to learn representations with various emergent properties (Schaul et al., 2015; Jaderberg et al., 2017; Machado et al., 2018; Bellemare et al., 2019; Gelada et al., 2019; Fedus et al., 2019; Dabney et al., 2021; Lyle et al., 2022). While most of these papers optimize auxiliary tasks in support of reward maximization from online interactions, our work investigates learning representations solely from auxiliary tasks on offline datasets. Closely related to our work is the study of random cumulants (Dabney et al., 2021; Lyle et al., 2021), both of which identify random cumulant auxiliary tasks as being especially useful in sparse-reward environments. Our work differs from these prior works in both motivation and implementation. Notably absent in prior work on random cumulants is the study of representational capacity as a function of the number of tasks.

Another body of related work on decoupling representation learning from RL primarily revolves around the use of contrastive learning (Anand et al., 2019; Wu et al., 2019; Stooke et al., 2021;

Schwarzer et al., 2021b; Erraqabi et al., 2022). Anand et al. (2019) proposed ST-DIM, a collection of temporal contrastive losses operating on image patches from environmental observations. Although the representations learned by ST-DIM are able to predict annotated state-variables in Atari 2600 games, their pretraining method was never evaluated for control. Stooke et al. (2021) uses contrastive learning for learning the temporal dynamics, resulting in minor improvements in online control from a fixed representation. Additionally, Schwarzer et al. (2021b) augments next-state prediction with goal-conditioned RL and inverse dynamics modelling, enabling strong performance on Atari 100k benchmark (Kaiser et al., 2020). Our work is complementary to these prior works and investigates the utility of scaling auxiliary tasks for learning good representations, which in principle can be easily combined with existing approaches. Additionally, recent work on using state-similarity metrics tackles the representation learning problem through the lens of behavioral similarity (Castro et al., 2021; Zhang et al., 2021; Agarwal et al., 2021a). We note that, in contrast to our method, the behavioral metrics used in these works are heavily based on the reward structure of the environment.

Related to our method, Touati & Ollivier (2021) consider representation learning with the successor measure (see also Touati, 2021, Algorithm 7). Algorithmically, their approach differs from ours in a number of ways, including the use of a learned state density function in lieu of indicator functions, the decomposition of the successor measure into its so-called forward and backward representations, and a bespoke sampling procedure to generate sample trajectories from which the representation is learned. By comparison, our approach directly constructs a relevant set of auxiliary tasks, which results in a significantly simpler algorithm that is more easily scaled according to available computational resources and to the full gamut of Atari 2600 games, as we will demonstrate.

Additionally, there has been recent work on framing the representation learning problem in RL as a min-max objective where you learn state features that can linearly represent, for example, a specific class of value-functions (Bellemare et al., 2019) or the Bellman backup itself (Modi et al., 2021; Zhang et al., 2022). Although we do not frame our method in terms of a min-max formulation, we do seek to learn a representation that can linearly predict the value function given any reward function. These previous works are primarily theoretical in nature, with some assuming specific structure of the MDP. In contrast, our class of auxiliary prediction tasks allows us to learn representations in environments with large, high-dimensional state-spaces without any of these prior assumptions.

## 3   Background

The RL problem can be modeled as a Markov Decision Process (MDP) defined by the 5-tuple $\mathcal{M} = \langle \mathcal{X}, \mathcal{A}, \mathcal{R}, \mathcal{P}, \gamma \rangle$, in which $\mathcal{X}$ is a set of states, $\mathcal{A}$ is a set of actions, $\mathcal{R} : \mathcal{X} \times \mathcal{A} \mapsto \mathbb{R}$ is a scalar reward function, $\mathcal{P} : \mathcal{X} \times \mathcal{A} \mapsto \mathscr{P}(\mathcal{X})$ is a transition function that maps state-action pairs to a distribution over next states, and $\gamma \in [0, 1)$ is a discount factor. A policy $\pi : \mathcal{X} \mapsto \mathscr{P}(\mathcal{A})$ is a function that maps states to a distribution over actions.

The goal of an RL agent is to learn a policy that maximizes the cumulative discounted rewards from the environment, also known as the discounted return. The state-action value function is defined as the expected discounted return when starting in a state and following the policy $\pi$:

$$Q^\pi(x, a) := \mathbb{E}_{\pi, \mathcal{P}} \left[ \sum_{t=0}^{\infty} \gamma^t \mathcal{R}(X_t, A_t) \mid X_0 = x, A_0 = a \right].$$

In this paper, we consider approximating the value function $Q^\pi$ using a linear combination of features. We call the map $\phi : \mathcal{X} \to \mathbb{R}^k$ a $k$-*dimensional state representation*; $\phi(x)$ is the feature vector for a state $x \in \mathcal{X}$. The value function approximant at $(x, a)$ is

$$\hat{Q}(x, a) = \phi(x)^\top w_a,$$

where $w_a \in \mathbb{R}^k$ is a weight vector associated with action $a$. In deep RL, the state representation is parameterized by a neural network. Often, the representation is learned end-to-end by optimizing the parameters to make more accurate predictions about the value function. Additional predictions that further shape the state representation are called *auxiliary tasks* (Jaderberg et al., 2017). In this work, we write $\mathcal{T}$ for the set of auxiliary tasks.

The *successor representation* (SR; Dayan, 1993) encodes the temporal structure of the MDP in terms of which states can be reached from any other state under a given policy. It is given by

$$\psi_{\text{SR}}^\pi(x, a, \tilde{x}) = \sum_{t=0}^{\infty} \gamma^t \mathbb{P}\{X_t = \tilde{x} \mid X_0 = x, A_0 = a, A_{t>0} \sim \pi\}.$$

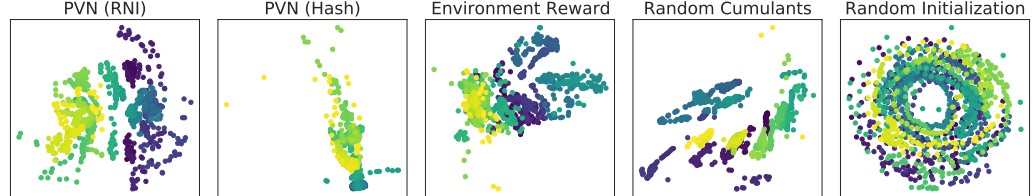

Figure 1: MDS plots of the features learned by PVN and other baseline methods. Each plotted point corresponds to a state in a single trajectory on the game CHOPPER COMMAND. The episode starts with the dark purple points and ends with the light yellow points. Environment Reward corresponds to features learned by optimizing the environment reward directly.

A convenient, recursive form expresses the SR in terms of an indicator function, highlighting that for each $\tilde{x}$, the SR is the value function associated with the reward function $\mathcal{R}(x,a) = \mathbf{1}\{x = \tilde{x}\}$:

$$\psi_{\text{SR}}^{\pi}(x, a, \tilde{x}) = \mathbf{1}\{x = \tilde{x}\} + \gamma \, \mathbb{E}_{\pi}\left[\psi_{\text{SR}}(X', A', \tilde{x}) \mid X = x, A = a\right].$$

## 4 Proto-value networks

In this section, we derive our *proto-value networks* algorithm. At a high level, this algorithm learns a state representation that approximates the singular vectors associated with the successor measure, the extension of the SR to continuous state spaces. We do this in order to derive an algorithm that is more suitably tailored to the large state spaces of deep RL domains, where many states are encountered once or never at all, and some notion of distance between states must be accounted for.

To gain some understanding into this process, let us consider how the method of auxiliary tasks (Jaderberg et al., 2017) can be used to obtain a state representation that approximates the SR. In the tabular setting, where $\mathcal{X}$ and $\mathcal{T}$ are of finite sizes $n$ and $m$ respectively, we write the feature matrix $\Phi \in \mathbb{R}^{n \times d}$, so that each state $x$ is associated with a feature vector $\phi(x) \in \mathbb{R}^d$. Given an auxiliary task matrix $\Psi \in \mathbb{R}^{n \times m}$, the method of auxiliary tasks can be shown to be equivalent to minimizing the loss function

$$\mathcal{L}(\Phi, W) = \|\Phi W - \Psi\|_F^2 = \sum_{x \in \mathcal{X}, i \in \mathcal{T}} \left(\phi(x)^{\top} w_i - \psi_i(x)\right)^2$$

jointly with respect to $\Phi$ and $W$. Here, $W \in \mathbb{R}^{d \times m}$ is a weight matrix with columns $(w_i)_{i=1}^m$ and $\psi_i(x)$ is the entry of $\Psi$ corresponding to state $x$ and task $i$. In the sequel, we will assume that a near-optimal $W$ can be obtained easily and simply consider the loss

$$\mathcal{L}(\Phi) = \min_W \mathcal{L}(\Phi, W),$$

to be minimized over $\Phi$. It is known (e.g., Bellemare et al., 2019) that any feature matrix that minimizes this loss function must have columns that lie in the subspace spanned by the top $d$ left singular vectors of $\Psi$. In particular, when $\Psi$ is square and symmetric the auxiliary task method recovers the subspace spanned by its top $d$ eigenvectors.

Here, we are interested in the setting in which $\Psi^{\pi_r}$ is the SR matrix for the *uniformly random policy*. In the symmetric case, the eigenvectors of $\Psi^{\pi_r}$ form what is called the *proto-value functions* of the MDP (Mahadevan & Maggioni, 2007). These eigenvectors are of special importance because they encode the spatial structure of the MDP in terms of a diffusion process, and have been shown to correlate with neural encodings of spatial location in mammals (Stachenfeld et al., 2014).

### 4.1 Extension to the random successor measure

Let $\pi$ be a policy and $\Sigma$ the power set of $\mathcal{X}$. The successor measure $\psi^{\pi} : \mathcal{X} \times \mathcal{A} \times \Sigma \to \mathbb{R}$ extends the SR to quantify the discounted visitation frequency of an agent, in expectation over trajectories

---

Behzadian & Petrik (2018) gives the singular-vector extension for the asymmetric case. Because this extension is straightforward and symmetry rarely holds, in this paper we use the term proto-value networks to describe state representations learned in both the symmetric and asymmetric settings.

and for *various subsets of the state space* (Blier et al., 2021). Given a set $S \subset \mathcal{X}$, we write

$$\psi^\pi(x, a, S) = \sum_{t=0}^\infty \gamma^t \mathbb{P}\{X_t \in S \mid X_0 = x, A_0 = a, A_{t>0} \sim \pi\}.$$

As with the SR, this can be expressed in terms of an expectation over an indicator function, and further decomposed in a Bellman equation:

$$\psi^\pi(x, a, S) = \sum_{t=0}^\infty \mathbb{E}_\pi \left[ \gamma^t \mathbf{1}\{X_t \in S\} \mid X_0 = x, A_0 = a \right]$$
$$= \mathbf{1}\{x \in S\} + \gamma \, \mathbb{E}_\pi \left[ \psi^\pi(X', A', S) \mid X = x, A = a \right].$$

The passage from state equality to set inclusion is particularly appealing in deep RL: first, because states rarely repeat along a trajectory or between episodes, the indicator $\mathbf{1}\{x = y\}$ is almost always zero. Second, set inclusion allows us to incorporate a notion of closeness to $\psi^\pi$, e.g. by focusing on subsets $S$ that include semantically similar states. We will return to this point later in the section.

By analogy with the tabular setting, let us now define a loss function which, if suitably minimized, should produce a useful state representation. For ease of exposition, we continue to assume that $\mathcal{X}$ is finite, although perhaps very large; the reader interested in a proper mathematical treatment of the full continuous-state setting is invited to consult Blier et al. (2021) and Pfau et al. (2019).

Let $\xi$ be a distribution over subsets of states and $\Xi \in \mathbb{R}^{n \times n}$ is a diagonal matrix with entries $\{\xi(x) : x \in \mathcal{X}\}$ on the diagonal. The *Monte Carlo successor measure loss* is

$$\mathcal{L}_{MCSM}(\Phi) = \min_{w_{S,a} \in \mathbb{R}^d} \mathop{\mathbb{E}}_{S \sim \xi} \left[ \Big( \sum_{x \in \mathcal{X}, a \in \mathcal{A}} (\phi(x)^\top w_{S,a} - \psi^\pi(x, a, S))^2 \right].$$

**Theorem 1.** *If $\Phi^*$ is a feature matrix minimizing $\mathcal{L}_{MCSM}(\Phi)$, then its column space spans the top $d$ left singular vectors of the (infinite-dimensional) successor measure matrix $\Psi^\pi$ with respect to the inner product $(x, y)_\Xi = y^\top \Xi x$, for all $x, y \in \mathbb{R}^n$.*

In practice, samples of $\psi^\pi(x, a, S)$ (which must be estimated from complete trajectories) are not available; instead, it is preferable to learn an approximation by bootstrapping (Sutton & Barto, 2018). The corresponding temporal-difference successor measure loss is

$$\min_{w_{S,a} \in \mathbb{R}^d} \mathop{\mathbb{E}}_{S \sim \xi} \left[ \Big( \sum_{x \in \mathcal{X}, a \in \mathcal{A}} (\mathbf{1}\{x \in S\} + \gamma \mathop{\mathbb{E}}_\pi \left[ \phi(X')^\top w_{S,A'} \mid X = x, A = a \right] - \phi(x)^\top w_{S,a})^2 \right]; \quad (1)$$

we will use this form in the derivations that follow.

## 4.2 A practical implementation

Our algorithm aims to approximate the loss in Equation 1 using tools from deep RL. We first approximate the expectation over $\xi$ by sampling a collection of sets $(S_i)_{i=1}^m$ from $\xi$. These sets are kept fixed throughout learning. With this in mind, each set corresponds to an indicator function that we treat as a binary reward function $r_i(x) = \mathbf{1}\{x \in S_i\}$. The actual auxiliary task is then the value function of the random policy associated with this reward.

Denote by $\hat{\psi}_i(x, a)$ the prediction made by our neural network for state $x$, action $a$, and the set $S_i$. Given a sample transition $(x, a, x')$, we define the sample target

$$r_i(x) + \gamma \frac{1}{|\mathcal{A}|} \sum_{a' \in \mathcal{A}} \hat{\psi}_i(x', a').$$

Notice that the average over the next-action $a'$ arises as a consequence of taking the policy $\pi$ to be uniformly random. We then train the neural network by performing stochastic gradient descent on the loss derived from this sample target:

$$\left( r_i(x) + \gamma \frac{1}{|\mathcal{A}|} \sum_{a' \in \mathcal{A}} \hat{\psi}_i(x', a') - \hat{\psi}_i(x, a) \right)^2.$$

Following common usage, the actual gradient estimate is obtained by aggregating multiple transitions into a minibatch and applying the Adam optimizer (Kingma & Ba, 2015).

Before explaining how the sets $S_i$ are defined, let us remark on a number of appealing properties of these auxiliary tasks, when viewed from a deep RL perspective. First, the use of a random policy means that learning usually proceeds in an off-policy manner. However, we expect this to be a relatively mild form of off-policy learning, one that is in general much more stable than one derived by maximization, as in a Bellman optimality equation. Although one could also learn the value function associated with the current policy (as in SARSA (Rummery & Niranjan, 1994)), this precludes the use of offline datasets for learning the representation, or at least makes the learned representation strongly dependent on the behaviour policy. By contrast, the representation learned by PVN only depends on the availability of data. In effect, these auxiliary tasks *depend only on the structure of the environment, and not on the agent's behaviour*.

We also expect binary reward functions to be easier to tune than, say, those derived from a distance function (dependent on getting the scale parameter correct) or real-valued random rewards (dependent on the underlying distribution). Binary rewards are particularly appealing in domains where the reward function is itself binary or ternary (i.e., Atari 2600 video games), in which case they can be adjusted to have similar statistics to the true reward function. We will demonstrate how to do this in the following section.

### 4.3 Generating indicator functions

Thus far we have described our algorithm as sampling sets of states $(S_i)_{i=1}^m$ which are then converted into a reward function by means of an indicator. In deep RL, this is inconvenient for two reasons: first, because it is not clear from what distribution of states should be sampled (how should one generate arbitrary video-game states?); second, because testing for set inclusion may also be brittle, effectively reducing to repeated equality tests. Instead, we opt here for an *implied set*, defined directly by its indicator function.

Let $\mathcal{F}$ be a family of functions mapping $\mathcal{X}$ to $\{0, 1\}$. Then, for any function $f \in \mathcal{F}$, its implied set is

$$S_f = \{x \in \mathcal{X} : f(x) = 1\}.$$

Of course, this is equivalent to

$$f(x) = \mathbf{1}\{x \in S_f\}.$$

Sampling functions from $\mathcal{F}$ according to some distribution $\xi_f$ and using them in lieu of the indicator is therefore equivalent to sampling sets of states for some distribution $\xi$ implied by $\xi_f$. The advantage is that testing for inclusion in $S_f$ only requires the evaluation of $f$ at $x$, which for carefully-chosen functions can be done at little computational cost.

The simplest scenario occurs when the family $\mathcal{F}$ is parametrized by some weight vector $\theta$, so that the random function $f_\theta$ corresponds to a random set of states. In this paper we consider two such families of functions: universal hash functions and random network indicators. Both families are *tunable*, in the sense that they are parametrized so that the implied sets $S_f$ each cover a desired fraction of the overall state space. In probabilistic terms, tunable means that we can with minimal or no computation find parameters such that for any given state $x$,

$$\mathbb{P}\{x \in S_f\} = p.$$

Here, the probabilistic statement is with respect to the draw of $f$ from $\mathcal{F}$. For universal hash functions, the tuning is immediate from the algorithm, and so we describe it first.

A *Carter-Wegman* family of hash functions $\mathcal{F}_{\text{CW}}$ (Carter & Wegman, 1979) consists of functions mapping each integer $x \in \mathbb{N}$ to the set $\{0, \ldots, k-1\}$, with the property that

$$\mathbb{P}\{h(x) = i\} = \tfrac{1}{k} \text{ for } i = 0, \ldots, k-1,$$

where the probabilistic statement is over the random draw of $h$ from $\mathcal{F}_{\text{CW}}$. One may think of a CW family as deterministically assigning labels to integers $x$ (in the sense that $f$ is deterministic), but randomly (in the sense that $f$ is random). See Appendix D.1 for full implementation details.

We construct our tunable indicator function as

$$f(x) = \mathbf{1}\{h(x) = 0\}.$$

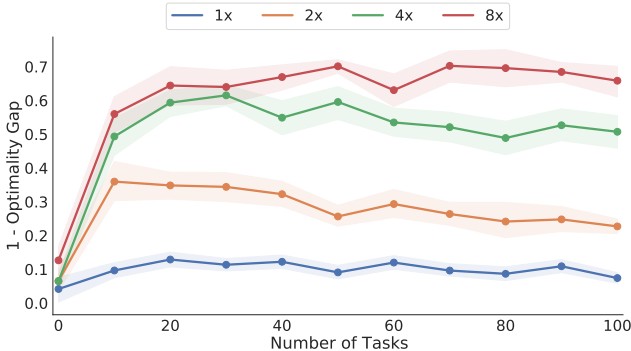

Figure 2: Performance of agents performing linear value approximation on top of a learned PVN. Agents are trained for 4 million environment frames; the legend indicates the network size as a capacity multiplier. Scaling network capacity with the number of auxiliary tasks leads to better performance. Larger networks support a larger number of auxiliary tasks.

By construction, choosing $k = \frac{1}{p}$ yields the desired tuning (up to integer rounding). In our setting, $x$ is a high-dimensional observation (for example, an image) rather than an integer; yet we will see that, perhaps surprisingly, encoding each image as a unique integer is sufficient to produce better-than-random state representations.

One drawback of using universal hash functions to define sets of interest is that they may assign different values to perceptually near-identical states (a single pixel difference suffices). Following common usage (Burda et al., 2019; Dabney et al., 2021), we may use randomly initialized neural networks to map similar states to similar values. Specifically, let us view a randomly initialized, single-output DQN network as a function $g : \mathcal{X} \to \mathbb{R}$. We further decompose this function into a map $g_1 : \mathcal{X} \to \mathbb{R}^l$ and a linear map from $\mathbb{R}^l \to \mathbb{R}$:

$$g(x) = g_1(x)^\top \omega + b,$$

where $\omega$ is a parameter vector and $b \in \mathbb{R}$ is a bias term. With this in mind, we may simply construct the indicator function

$$f(x) = \mathbf{1}\{g(x) \geqslant 0\}.$$

The result, however, is not yet tunable: it is hard to choose the right distribution of network weights so that a desired fraction of states satisfy $f(x) = 1$. However, for any $p \in [0, 1]$ and any non-zero fixed $\omega$, $g_1$, and distribution of states $\mu$, there exists a bias term $b'$ such that

$$\mathbb{P}_{x \sim \mu}\{g_1(x)^\top \omega + b' \geqslant 0\} = p\,.$$

Such a bias term can accurately be determined from a small number of online interactions using the method of quantile regression (Koencker, 2005); the exact update rule is given in Appendix D.2. With this method, we obtain network-derived indicator functions that are tunable and are likely to assign similar values to perceptually similar states. We refer to this class of indicator functions as *random network indicators* (RNIs), which we empirically evaluate in the following section.

## 5    Empirical Analysis

To disentangle the contributions of the primary and auxiliary tasks on the expressiveness of the learned features, we split our learning procedure in two parts: a representation pre-training phase, and an online RL phase. During the representation pre-training phase, we use transition data from offline Atari datasets in RL Unplugged (Agarwal et al., 2020; Gulcehre et al., 2020) and the procedure described in Section 4 to train an encoder which acts as a feature extractor (see Appendix D for complete implementation details). Note that while this dataset contains environment rewards, none of the methods make use of the environment rewards unless explicitly stated. Following the pre-training phase, we fix the weights of the learned encoder and train an RL agent online directly from this "frozen" representation. Notably, we train for *only 3.75 million* agent steps, compared to the 50 million agent steps (200M Atari 2600 frames) that is standard in most Atari setups. Our agents are

---

This corresponds to using a SIGN nonlinearity at the end of the network.

implemented using the Acme library (Hoffman et al., 2020). Our hyperparameter choices for both phases of training can be found in Appendix D.5.

## 5.1 Scaling capacity with auxiliary tasks

Prior work indicates that the optimal number of auxiliary tasks for representation learning is unexpectedly small, and that scaling up the number of auxiliary tasks can hurt performance (Lyle et al., 2021, Fig. 5). We expect that the representational capacity of the neural network has a strong effect on the number of auxiliary tasks we are able to learn with. To study this effect, we use the Impala-CNN network (Espeholt et al., 2018) and vary its effective width; that is, we multiply the number of convolutional filters and the number of features in the penultimate layer. We select a width multiplier in the set $\{1, 2, 4, 8\}$ and sweep the number of tasks from $\{0, 10, \dots, 100\}$. For this experiment, we use 5 games (ASTERIX, BEAM RIDER, PONG, QBERT, and SPACE INVADERS) with 3 seeds for pre-training, resulting in 15 encoders for each combination of width multiplier and number of auxiliary tasks. During the online phase, we train with 3 seeds per encoder, resulting in a total of 45 runs per sweep configuration. We evaluate for 100 episodes after 1M agent steps.

We summarize our results using Rliable (Agarwal et al., 2021b). Figure 2 depicts the optimality gap (distance from human-level performance). We find that increasing the representational capacity of the network increases performance, even for a very small number of tasks. This is perhaps surprising, since it indicates that we only need a handful of tasks to train large-scale representations, corroborating results by Lyle et al. (2021). Though a small part of this performance gain might be obtained just by virtue of having more output features (following the lottery ticket hypothesis (Frankle & Carbin, 2019)), we can see that there is a marked improvement when we increase the number of auxiliary tasks from 0 for all network sizes.

We further find that as network capacity is increased, the algorithm can use more auxiliary tasks to improve its representation. For example, while the $2\times$ network achieves maximal performance with 10 tasks, the $8\times$ network performs best in the range of $[50, 100]$ tasks. This gives evidence for the scalability of PVN as an approach for learning rich state representations.

## 5.2 Evaluating the learned representation

Using the insights gained from our scaling experiment, we evaluate a model with a large number of auxiliary tasks on a broader suite of Atari games. We use the $8\times$ network and fix the number of auxiliary tasks to 100, which empirically performed well. We use the same training setup described in the previous section, though we use all 46 games available in RL Unplugged. We use 3 seeds for offline pre-training, and 3 additional seeds per encoder during online training. We train for 3.75M agent steps, and evaluate for 100 episodes. We compare against the following pre-training baselines:

**Random Initialization**: Randomly initialized features using the same network architecture. This simple baseline should confirm that the efficacy of our representations come from our pre-training procedure, and not merely because we use a large encoder network.

**Random Cumulants (RCs)**: Random reward functions introduced by Dabney et al. (2021), and later expanded upon by Lyle et al. (2021). This method is similar to ours, but uses a random reward $r_i(x, x') = s \cdot (f(x') - f(x))$ instead of the random indicator function, and replaces the average over next-state actions by a maximization (off-policy learning of the optimal policy for each cumulant). Here, $f$ is also given by a random network.

**Self-Predictive Representations (SPR)**: A contrastive-learning method that directly optimizes for temporal consistency of the learned representation (Schwarzer et al., 2021a). It does so by learning a latent-space transition model and forcing subsequent states to have similar representations.

**Behavior Cloning (BC)**: Behavior Cloning has been shown as a strong baseline in Offline RL, especially when increasing the amount of pre-training data (Schwarzer et al., 2021b; Baker et al., 2022). It should give a strong indication of the performance that is possible when using large datasets.

For each of these methods, we freeze the 8x encoder after the pre-training stage and use the previously-described online training scheme. Figure 3 illustrates that PVN outperforms these baselines in all aggregate metrics. We also note that PVN using linear function approximation (3.75M agent interactions) is competitive with DQN (50M agent interactions) in many games, as illustrated in the per-game results found in Appendix F.

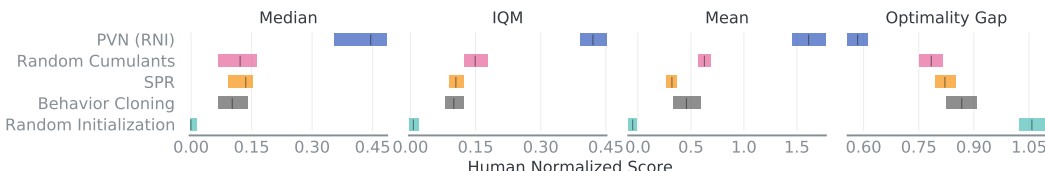

Figure 3: Performance of PVN RNI vs other methods described in Section 5.2. Computed using 125 seeds, aggregated across 46 Atari games.

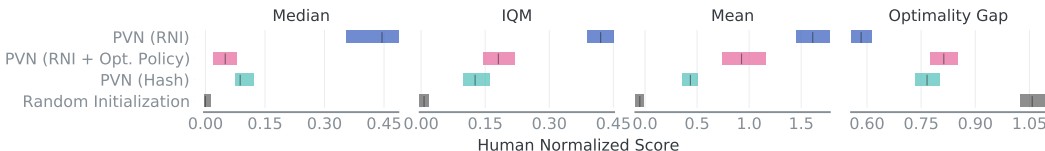

Figure 4: **PVN (Hash)**: PVN with hash indicator functions. **PVN (RNI)**: PVN with random network indicators. A randomly initialized network with $(8\times)$ capacity is plotted for comparison.

We visualize the learned representations from different methods using multidimensional scaling (MDS) plots in Figure 1 (with more games in Appendix E). These plots show that different methods clearly lead to representations with different structures. Notably, the representations learned by PVN (RNI) place temporally-successive states close together, and appears to capture information about the dynamics of the environment without requiring access to the environment reward.

### 5.3 Ablations

We perform ablative experiments to verify the importance of the different PVN components. First, we validate our choice of indicator function by replacing RNIs with the hash indicator functions described in Section 4. We compare their performance in Figure 4, which shows that hash indicator functions perform poorly compared to RNIs; this indicates that the choice of indicator function is an important design decision. We expect that the inductive biases in random convolutional networks allow RNIs to include a notion of state similarity in the tasks they induce.

Next, we hypothesize that using the random policy as the target policy is a key contributor to PVN's performance. To verify this hypothesis, we ablate the TD-target of our learning update to maximize over the next-state action-values, as per the Bellman optimality equation. When the mean function is replaced with the max function in the TD backup, PVN attempts to learn the optimal value function for each indicator function, rather than the value function of the random policy. The result of this experiment can be seen in Figure 4. Using the mean formulation has a much higher median human normalized score than the max formulation. This is likely due to instability that arises from max bias and state coverage due to the off-policy learning required for the optimal value function. Learning the value function of the random policy also requires off-policy learning; however, we predict that it doesn't have such a large effect, as we previously described in Section 4.2.

## 6  Discussion

While our experiments have shed some light on the scalability of auxiliary tasks, there are a number of remaining open questions that represent exciting opportunities for further exploration. An exciting future direction is to use insights from the literature on scaling models effectively (Tan & Le, 2019) to further scale the auxiliary tasks we introduced here. Orthogonally, it may still be possible to train with more tasks without increasing the capacity of the network. It is surprising that with even a relatively large network with tens of millions of parameters, such as Impala $(8\times)$, the network only supports a handful of tasks. It is not clear why training with more tasks leads to worse performance, especially for smaller Impala architectures. Finally, in line with Agarwal et al. (2022), we will open-source our pre-trained representations, which we hope would enable researchers to tackle credit assignment on ALE, without the excessive computational cost of re-learning such representations.

## Acknowledgements

We thank Nathan Rahn, Max Schwarzer, Harley Wiltzer, Wesley Chung, Adrien Ali Taïga, David Meger, and Doina Precup for their useful feedback on this work. A special thanks to Wesley Chung for completing the tabular proof presented in Appendix C. This work was supported by the National Sciences and Engineering Research Council of Canada (NSERC) and the Canada CIFAR AI Chair program.

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

# A  Background

## A.1  Proto-Value Functions

Proto-Value Functions on Four-Room Grid

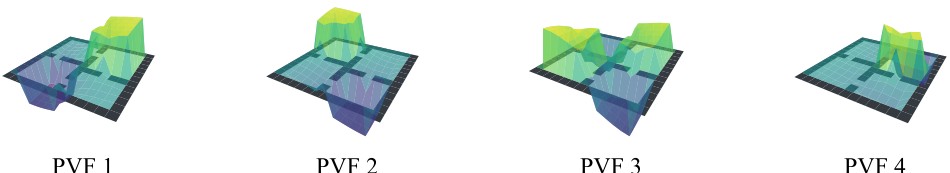

PVF 1       PVF 2       PVF 3       PVF 4

Figure 5: First four proto-value functions (eigenvectors of $\Psi^\pi$ when $\pi$ is the uniform random policy) on the Four Room grid world.

Proto-Value Functions (PVFs) are defined in terms of the graph Laplacian $L \in \mathbb{R}^{n \times n}$, that is

$$L = D - A,$$

where $D \in \mathbb{R}^{n \times n}$ is the degree matrix and $A \in \mathbb{R}^{n \times n}$ is the adjacency matrix. The actual PVFs are defined as the eigenvectors of the graph Laplacian, that is the non zero vectors $v \in \mathbb{R}^n \setminus \{0\}$ verifying

$$Lv = \lambda v.$$

where $\lambda \in \mathbb{R}$ is the eigenvalue associated with the eigenvector $v$.

Individually, these eigenvectors correspond to different time-scales of the diffusion process of a random-walk over the state-space (Mahadevan & Maggioni, 2007). Intuitively, PVFs can be thought of as capturing large-scale temporal properties of the environment. Figure 5 shows an example of the first four PVFs on the Four-Room domain (Sutton et al., 1999; Solway et al., 2014) to give some intuition for their structure.

## A.2  The Successor Representation

Let $P^\pi \in \mathbb{R}^{n \times n}$ be the transition matrix and $r^\pi$ the reward vector, both induced by the policy $\pi$. We can now write the policy evaluation equation for the values $v^\pi \in \mathbb{R}^n$ as:

$$v^\pi = \underbrace{(I - \gamma P^\pi)^{-1}}_{\Psi^\pi} r^\pi,$$

where $\Psi^\pi$ is the Successor Representation (SR). We can also write each element of the SR as the expected discounted future occupancy for a state $s'$ given you start in a state $s$:

$$\Psi^\pi(s, s') = \sum_{t>0} \gamma^t \mathbb{P}(S_t = s' \,|\, S_0 = s)$$
$$= \mathbb{E}_\pi \left[ \gamma^t \mathbf{1}\left\{S_t = s'\right\} \,|\, S_0 = s \right].$$

## A.3  Connection Between the SR & PVFs

We can further connect the Successor Representation with Proto-Value Functions under some assumptions.

**Assumption 1.** *The Successor Representation is defined with respect to the uniform random policy.*

**Assumption 2.** *The transition matrix $P^\pi$ is symmetric.*

Under the above assumptions, we have that the eigenvectors of $\Psi^\pi$ are equivalent to the PVFs (eigenvectors of $L$) (Machado et al., 2017; Stachenfeld et al., 2014). This helps motivate the choice of the uniform random policy as the target policy in the PVN TD update.

## B Proofs for Section 4

**Theorem 1.** *If $\Phi^*$ is a feature matrix minimizing $\mathcal{L}_{MCSM}(\Phi)$, then its column space spans the top $d$ left singular vectors of the (infinite-dimensional) successor measure matrix $\Psi^\pi$ with respect to the inner product $(x, y)_\Xi = y^\top \Xi x$, for all $x, y \in \mathbb{R}^n$.*

*Proof.* We consider the SVD of the successor measure $\psi$ with respect to the weighted inner product $\Xi$. In matrix form, we write

$$\Psi = F \Sigma B^\mathsf{T}$$

where $F \in \mathbb{R}^{n \times d}, \Sigma \in \mathbb{R}^{d \times d}$ and $B \in \mathbb{R}^{n \times d}$ satisfy

$$F^\mathsf{T} F = I, B^\mathsf{T} \Xi B = I, \Sigma = \mathrm{diag}(\sigma_1, ..., \sigma_d)$$

and $\sigma_i$ are the singular values of $\Psi$ sorted in decreasing order.

$$\begin{aligned}
\arg\min_{\Phi \in \mathbb{R}^{n \times d}} \mathcal{L}_{MCSM}(\Phi) &= \arg\min_{\Phi \in \mathbb{R}^{n \times d}} \min_{w_{S,a} \in \mathbb{R}^d} \mathbb{E}_{S \sim \xi} \left[ \Big( \sum_{x \in \mathcal{X}, a \in \mathcal{A}} (\phi(x)^\top w_{S,a} - \psi(x, a, S))^2 \Big) \right] \\
&= \arg\min_{\Phi \in \mathbb{R}^{n \times d}} \min_{W} \| (\Phi W - \Psi) \|_\Xi^2 \\
&= \arg\min_{\Phi \in \mathbb{R}^{n \times d}} \| \Pi_\Phi^\perp \Psi \|_\Xi^2
\end{aligned}$$

where $\Pi_\Phi$ is the orthogonal projection onto $\mathrm{span}(\Phi)$. The above is equivalent to saying that $\Phi$ must span the top $d$ singular vectors of $\Psi$. $\square$

## C Tabular Results

Define the Successor Representation (SR) as $\Psi^\pi = (I - \gamma P^\pi)^{-1} \in \mathbb{R}^{n \times n}$ and assume that $P^\pi$ is symmetric. Let $\mathcal{G}_k \in \mathbb{R}^{n \times \binom{n}{k}}$ be the matrix containing all the binary vectors corresponding to all $\binom{n}{k}$ subsets (i.e., its columns have all possible k-hot binary vectors). For example, given $n = 4$ we have,

$$\mathcal{G}_2 = \begin{bmatrix} 1 & 1 & 1 & 0 & 0 & 0 \\ 1 & 0 & 0 & 1 & 1 & 0 \\ 0 & 1 & 0 & 1 & 0 & 1 \\ 0 & 0 & 1 & 0 & 1 & 1 \end{bmatrix}.$$

In the tabular setting we seek to learn the successor measure with respect to $\mathcal{G}_k$ by minimizing

$$\mathcal{L}(\Phi, W) = \| \Phi W - \Psi \mathcal{G}_k \|_F^2.$$

We know that the optimal $\Phi$ will span the principal components of $\Psi \mathcal{G}_k$ (Bellemare et al., 2019). Note that when $k = 1$ we have, $\Psi \mathcal{G}_1 = \Psi$ in which case the principal components are the PVFs Machado et al. (2017). We want to characterize the principal subspace of $\Psi \mathcal{G}_k$ for $1 < k < n$.

**Claim**: $C = (\Psi \mathcal{G}_k)(\Psi \mathcal{G}_k)^\top$ has the same eigenvectors for all $k \in \{1, \ldots, n - 1\}$.

**Proof**: We start by writing down the covariance matrix as

$$\begin{aligned}
C &= (\Psi \mathcal{G}_k)(\Psi \mathcal{G}_k)^\top \\
&= \Psi \mathcal{G}_k \mathcal{G}_k^\top \Psi^\top.
\end{aligned}$$

The matrix $\mathcal{G}_k \mathcal{G}_k^\top$ is a double-constant matrix (O'Neill, 2021), i.e., it has a constant $a$ on the diagonal and a constant different from $a$ on the off-diagonal:

$$M_k = \mathcal{G}_k \mathcal{G}_k^\top = \begin{bmatrix} a & t & t & \cdots & t \\ t & a & t & \cdots & t \\ t & t & a & \cdots & t \\ \vdots & \vdots & \vdots & \ddots & \vdots \\ t & t & t & \cdots & a \end{bmatrix}.$$

In our case we have $a = \binom{n-1}{k-1}$ and $t = \binom{n-2}{k-2}$. Furthermore, we can use another property of double-constant matrices, we have that the eigenvalues of $M_k$ are $\lambda_1 = \lambda_{**} = a - t + n \cdot t$ and $\lambda_i = \lambda_* = a - t$ for all $i = 2, \ldots, n$. The eigenvectors for $\lambda_{**}$ are $v_{**} \propto \mathbf{1}$ where $\mathbf{1}$ is the vector of all ones. The eigenvectors for $\lambda_*$ are any non-zero vectors $v_*$ where $v_* \cdot \mathbf{1} = 0$, i.e., $v_*$ is orthogonal to the vector of all ones.

Next, we characterize the eigenspace of the matrix $\Psi^\pi$. We have,

$$
\begin{aligned}
\Psi &= (I - \gamma P^\pi)^{-1} \\
&= \left(I - \gamma Q \Lambda Q^{-1}\right)^{-1} & \text{(Since $P^\pi$ is symmetric, hence diagonalizable)} \\
&= Q \left(I - \gamma \Lambda\right)^{-1} Q^{-1}.
\end{aligned}
$$

This means that the eigenvectors of $\Psi^\pi$ are the same as the eigenvectors of $P^\pi$. We will denote the eigenvalues of $P^\pi$ to be $\lambda_i$ with associated eigenvectors $x_i$. For simplicity, we denote the eigenvalues of $\Psi^\pi$ as $\mu_i$ for $i = 1, \ldots, n$. Note that $\mu_i = (1 - \gamma \lambda_i)^{-1}$ for $i = 1, \ldots, n$.

Furthermore, since $P^\pi$ is a stochastic matrix, we have that $\mathbf{1}$ is an eigenvector with eigenvalue $1$. We let $x_1 = \mathbf{1}$ without loss of generality. Also, since $P^\pi$ is assumed to be symmetric, the eigenvectors can be chosen to be orthogonal to each other.

Putting this all together, take $x_i$ to be the $i$-th eigenvector of $\Psi^\pi$ (and $P^\pi$). We now have,

$$
\begin{aligned}
C x_i &= \Psi M_k \Psi^\top x_i \\
&= \Psi M_k \Psi x_i & \text{(by symmetry)} \\
&= \Psi M_k \mu_i x_i. & \text{($x_i$ is an eigevector of $\Psi$)}
\end{aligned}
$$

Now there are two cases:

*Case 1*: If $x_i = \mathbf{1}$ (and $i = 1$) we have,

$$
\begin{aligned}
C x_i &= \Psi M_k \mu_i x_i \\
&= \Psi \lambda_{**} \mu_1 \mathbf{1} \\
&= \mu_1 \lambda_{**} \mu_1 \mathbf{1} \\
&= \mu_1^2 \lambda_{**} \mathbf{1}
\end{aligned}
$$

*Case 2*: If $x_i \neq \mathbf{1}$ (and $i > 1$) we know that $x_i$ is orthogonal to $\mathbf{1}$ (since $P^\pi$ is a symmetric matrix) thus lies in the second eigenspace of $M_k$ corresponding to the eigenvector $v_*$. Therefore, we have,

$$
\begin{aligned}
C x_i &= \Psi M_k \mu_i x_i \\
&= \Psi \lambda_* \mu_i x_i \\
&= \mu_i \lambda_* \mu_i x_i \\
&= \mu_i^2 \lambda_* x_i.
\end{aligned}
$$

Thus, we have shown that $C = \Psi M_k \Psi^\top$ has the same eigenvectors as $\Psi$ and are independent of $k$. The new eigenvalues are $\mu_1^2 \lambda_{**}$ for the eigenvector $\mathbf{1}$ and $\mu_i^2 \lambda_*$ for all other eigenvectors $x_i$ for $i = 2, \ldots, n$.

## D   Implementation Details

### D.1   Universal Hash Functions

We define the set of multiply-shift universal hash functions (Carter & Wegman, 1979) as:

$$
h_i(x) = \left( a_0^{(i)} + \sum_{j=1}^{n} a_j^{(i)} \cdot x_j \mod p \right) \mod m,
$$

where $x \in \mathbb{R}^n$ is a flattened vector of the environment's observation, $a^{(i)} \in \mathbb{R}^n$ is a randomly initialized vector that that parameterizes the hash function, $p$ is a prime, which in our case is the Mersenne prime $p = 2^{13} - 1$, and $m$ allows us to control the activation proportion of the indicator function. We can now define the indicator function as follows:

$$
f_i(x) = \mathbf{1}\{h_i(x) = 0\}.
$$

## D.2 Quantile Regression

We use quantile regression to tune the proportion of states that trigger our random network indicator functions. To do so, we use a tunable bias that we update with gradient descent. First, recall that the random network indicators are computed using a random neural network $f : x \mapsto \mathbb{R}$. If we naively apply the SIGN function to the network output, the proportion of states that map to $1$ is unlikely to match the target proportion $p$. Therefore, we first add a bias term to the output $r' = f(x) + b$, and then tune the bias to minimize the quantile regression loss

$$\mathcal{L}_{\text{QR}}(b) = \mathbb{E}_{x \in \mathcal{X}} r'(x) \cdot ((1 - p) - \text{SIGN}(r'(x))) \tag{2}$$

Once the bias has been tuned, the output of the random network indicator is $r = \text{SIGN}(f(x) + b)$.

## D.3 Algorithm

Algorithm 1 gives pseudo-code for the method as implemented with a fixed replay memory.

---

**Algorithm 1** Proto-Value Networks

---

**Require:** Transition dataset $\mathcal{D}$, Function approximator $\hat{\Psi}_\theta : \mathcal{X} \to \mathbb{R}^{m \times |\mathcal{A}|}$, $m$ RNI networks $f_i : \mathcal{X} \to \mathbb{R}$, $m$ RNI threshold bias vectors $b_i$, Polyak coefficient $\tau$, reward proportion $p$

1: **for** step $= 1, \ldots$ **do**
2:     Sample mini-batch of $n$ transitions $\{(x, a, x')\}_{i=1}^n \subset \mathcal{D}$
3:
4:     *# Calculate random network indicators*
5:     $r'_j(x) \leftarrow f_j(x) + b_j \quad \forall j = 1, \ldots, m$
6:     $r_j(x) \leftarrow \text{SIGN}(r'_j(x))$
7:
8:     *# Calculate PVN loss*
9:     $\mathcal{L}_{\text{PVN}}(\theta) \leftarrow \frac{1}{n} \sum_{i=1}^n \frac{1}{m} \sum_{j=1}^m \left( r_j(x_i) + \gamma \frac{1}{|\mathcal{A}|} \sum_{a' \in \mathcal{A}} \hat{\Psi}_{\theta^-}^{(j)}(x'_i, a') - \hat{\Psi}_\theta^{(j)}(x_i, a_i) \right)^2$
10:
11:     *# Calculate quantile regression loss*
12:     $\mathcal{L}_{\text{QR}}(b_j) \leftarrow \frac{1}{n} \sum_{i=1}^n r'_j(x) \cdot ((1 - p) - \text{SIGN}(r'_j(x)))$
13:
14:     *# Perform gradient step*
15:     Update $\theta \leftarrow \theta - \eta_1 \frac{\partial}{\partial \theta} \mathcal{L}_{\text{PVN}}(\theta)$
16:     Update $b_j \leftarrow b_j - \eta_2 \frac{d}{db_j} \mathcal{L}_{\text{QR}}(b_j) \quad \forall j = 1, \ldots, m$
17:
18:     *# Polyak average target network parameters*
19:     $\theta^- \leftarrow \tau \theta^- + (1 - \tau) \theta$
20: **end for**

---

## D.4 Self-Predictive Representations (SPR)

We implement an 8x version of SPR (Schwarzer et al., 2021a) using the same parameters as in, Schwarzer et al. (2021a) except we take the final fixed representation to be the projection layer in addition to the convolutional encoder. This was done to maintain the number of features for all our pre-trained methods. We also train SPR for much longer than in the original paper, specifically, we perform the same number of gradient steps as PVN.

### D.5 Hyperparemeters

In the tables below we report all relevant hyperparameter choices for both our offline pre-training phase, and online learning phase.

We selected most of our hyperparameters based on best practices from previous work. We chose $p$ based on the estimated reward proportion from actual Atari games. We tuned our online hyperparameters using 5 tuning games, ASTERIX, BEAM RIDER, PONG, QBERT, and SPACE INVADERS.

Table 1: PVN Hyperparameters

| Hyperparameter | Value |
|---|---|
| Number of auxiliary tasks | 100 |
| Batch size | 256 |
| Number of gradient steps | 1,562,500 |
| Discount factor $\gamma$ | 0.99 |
| Target EMA coefficient $\tau$ | 0.99 |
| Reward proportion $p$ | 0.01 |
| Quantile regression burn-in steps | 62,500 |
| Tasks per module | 10 |
| Optimizer | Adam |
| Adam Learning rate | 1e-4 |
| Adam $\beta_1$ | 0.9 |
| Adam $\beta_2$ | 0.999 |
| Adam $\epsilon$ | 1.5e-4 |

Table 2: Online Hyperparameters

| Hyperparameter | Value |
|---|---|
| Update rule | DQN |
| Number of layers | 1 (linear) |
| Number of agent steps | 3.75M |
| Frame skip | 4 |
| Total frames | 15M |
| Train $\epsilon$ | 0.01 |
| Evaluation $\epsilon$ | 0.001 |
| n step | 1 |
| Discount factor $\gamma$ | 0.99 |
| Optimizer | Adam |
| Adam Learning rate | 6.25e-5 |
| Adam $\epsilon$ | 1.5e-4 |
| Maximum gradient norm | 10 |
| Batch size | 32 |
| Minimum replay size | 2,000 |
| Maximum replay size | 1,000,000 |
| Gradient updates per agent step | 1 |

# E    MDS Plots

Below are a selection of MDS plots for the methods discussed in the paper for each of the 5 tuning games. These plots are generated using the representations learned during the pre-training phase, and one expert trajectory is presented in each plot. Darker points correspond to states at the beginning of the trajectory, and lighter points correspond to states at the end of the trajectory. These plots demonstrate that the representations learned by each method are clearly different, and therefore have different properties. With these MDS explorations, we hope to gain some insight into the properties of each learnt representation. Motivated by PVFs, we expect a good (general) representation to capture the structure of the underlying transition dynamics of the environment. We note that PVN captures the temporal structure of each episode relatively well. With PVN, states that are near together in time have similar features, aligning with the properties of PVFs in the tabular case.

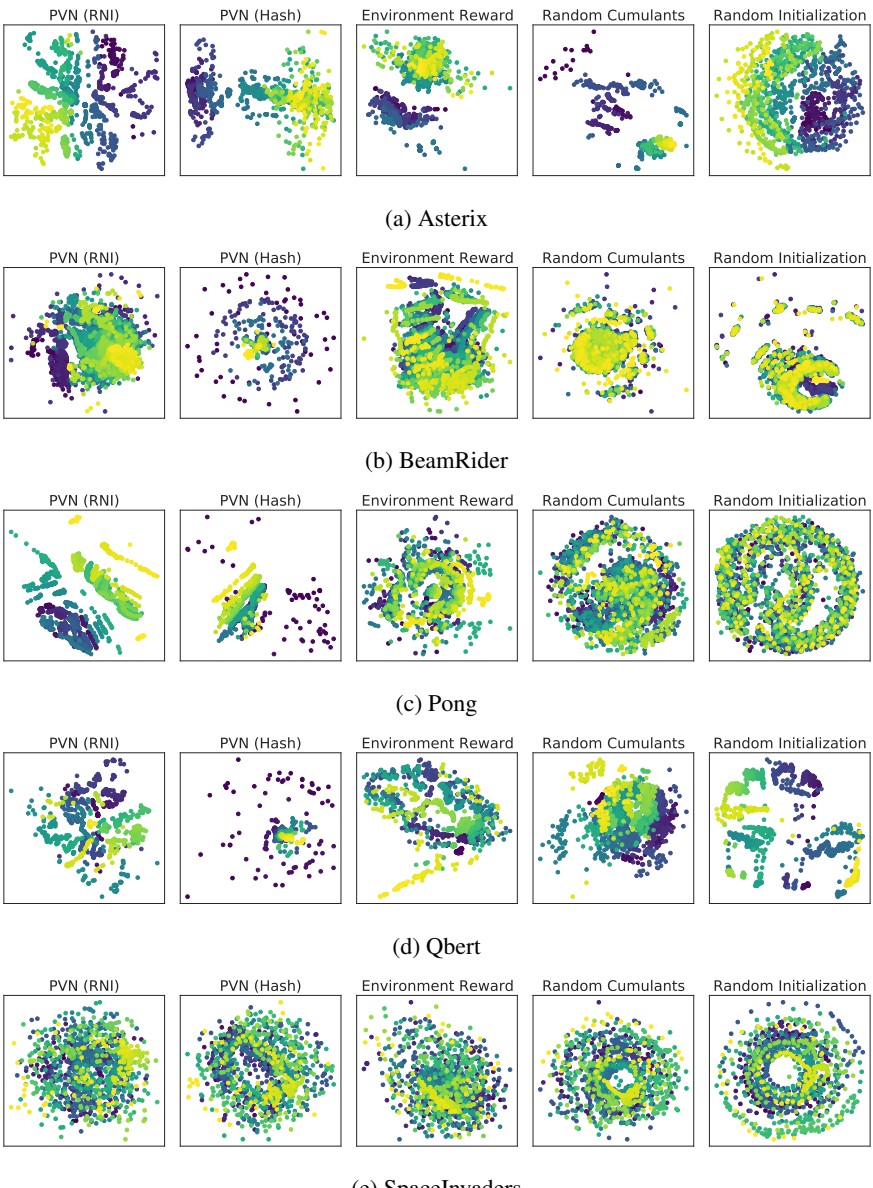

Figure 6: MDS plot for a single trajectory.

# F  Per-Game Results

Below, we report the per-game results for the methods discussed in the paper. In addition, we include DQN and the Environment Reward method, which trains an encoder using the environment reward during the pre-training phase, and then uses the fixed representation to train a linear head in the same manner as the compared methods. Note that Environment Reward acts as a kind of oracle in our setting – it is the only method that has access to the environment reward during the pre-training phase. The results reported here use 1 offline seed and 3 online seeds, and evaluation scores (averaged over 100 evaluation runs) are reported after 3.75M agent steps. DQN's performance is reported after 50M agent steps.

Table 3: Per-Game Results on RLU Games

| Game | DQN | Environment Reward | Random Initialization | Behavior Cloning | SPR | Random Cumulants | PVN (RNI) |
|---|---|---|---|---|---|---|---|
| Alien | 1,932.2 | 2,681.1 | 403.7 | 414.8 | 962.8 | 104.7 | **1,042.8** |
| Amidar | 893.7 | 169.1 | 43.8 | 50.2 | 46.8 | 43.8 | **63.6** |
| Assault | 1,442.5 | 695.1 | 193.2 | 452.9 | 293.0 | 494.6 | **1,100.7** |
| Asterix | 2,953.4 | 13,096.3 | 522.9 | 7,139.8 | 1,625.1 | 347.8 | **15,401.2** |
| Atlantis | 645,494.5 | 592,372.5 | 14,269.8 | 12,682.1 | 10,275.0 | **34,212.2** | 12,760.3 |
| BankHeist | 589.8 | 47.6 | 7.0 | 54.7 | 618.8 | **1,036.4** | 456.1 |
| BattleZone | 14,818.1 | 37,796.3 | 1,547.4 | 12,545.5 | 2,360.1 | 6,492.2 | **13,528.7** |
| BeamRider | 4,569.7 | 10,660.3 | 411.8 | 3,862.2 | 1,111.5 | 5,447.1 | **8,646.0** |
| Boxing | 74.7 | 99.5 | -30.2 | -4.7 | 53.0 | -2.8 | **87.2** |
| Breakout | 96.4 | 296.8 | 6.5 | 3.8 | 7.1 | **334.0** | 16.4 |
| Carnival | 4,438.2 | 4,976.0 | 362.2 | 671.0 | 684.8 | 648.9 | **1,228.5** |
| Centipede | 2,358.9 | 1,176.9 | 3,534.5 | 1,688.6 | **4,300.3** | 1,082.6 | 3,420.6 |
| ChopperCommand | 1,958.7 | 3,685.8 | 494.2 | 643.2 | 880.6 | 1,078.0 | **2,018.5** |
| CrazyClimber | 100,158.8 | 137,240.0 | 10,179.4 | 2,914.7 | 51,999.3 | **133,041.2** | 19,259.1 |
| DemonAttack | 4,427.9 | 101,850.7 | 123.1 | 14,426.7 | 421.8 | 2,407.8 | **78,671.1** |
| DoubleDunk | -14.2 | -18.5 | -19.6 | -18.4 | **-16.9** | -17.0 | -20.4 |
| Enduro | 668.4 | 1,593.9 | 22.7 | 126.8 | **509.4** | 14.7 | 426.4 |
| FishingDerby | -1.1 | 28.7 | -96.6 | -92.6 | -89.2 | -95.6 | **-72.5** |
| Freeway | 24.1 | 33.7 | 8.5 | 14.7 | 15.9 | **29.5** | 18.9 |
| Frostbite | 623.0 | 4,386.1 | 39.2 | 96.0 | **834.0** | 56.2 | 408.3 |
| Gopher | 4,579.2 | 1,114.5 | 231.1 | 479.8 | 939.6 | 1,459.0 | **3,070.0** |
| Gravitar | 214.4 | 1,481.9 | 42.7 | **310.0** | 72.8 | 77.6 | 164.3 |
| Hero | 12,348.1 | 9,932.4 | 544.5 | 1,974.9 | **9,256.3** | 445.5 | 1,942.2 |
| IceHockey | -7.3 | 21.8 | -13.2 | -9.1 | -14.9 | -12.2 | **-8.9** |
| Jamesbond | 473.3 | 1,058.6 | 24.4 | 387.9 | 82.2 | 69.7 | **630.2** |
| Kangaroo | 8,653.0 | 9,612.0 | 199.1 | **1,818.0** | 78.7 | 1,033.1 | 1,724.4 |
| Krull | 5,892.5 | 8,630.4 | 1,248.9 | 3,622.1 | 3,836.1 | 206.4 | **4,006.5** |
| KungFuMaster | 20,245.1 | 35,076.0 | 2,806.7 | 9,463.0 | **16,877.5** | 12,262.9 | 13,628.4 |
| MsPacman | 2,880.5 | 5,797.4 | 836.4 | 629.4 | **1,946.6** | 1,444.4 | 1,373.9 |
| NameThisGame | 6,114.5 | 17,612.3 | 1,425.9 | 2,519.8 | 2,036.0 | 2,418.6 | **7,218.7** |
| Phoenix | 4,764.3 | 24,501.3 | 414.6 | 904.1 | 935.8 | **7,828.6** | 6,946.1 |
| Pong | 11.5 | 21.0 | -20.8 | -14.3 | -14.0 | 6.8 | **20.1** |
| Pooyan | 3,114.5 | 1,675.0 | 747.7 | 444.9 | 829.5 | **1,443.3** | 1,432.2 |
| Qbert | 8,309.5 | 14,393.7 | 172.1 | 664.6 | 567.9 | 2,611.3 | **3,503.1** |
| Riverraid | 9,703.1 | 11,319.0 | 1,279.7 | 2,829.2 | 3,053.1 | 619.9 | **6,841.6** |
| RoadRunner | 36,386.9 | 37,417.8 | 2,543.2 | 939.2 | 3,932.6 | 418.0 | **6,975.3** |
| Robotank | 39.2 | 75.1 | 2.4 | **44.7** | 3.9 | 14.1 | 3.5 |
| Seaquest | 1,510.1 | 173.7 | 63.8 | 294.2 | 310.3 | 51.6 | **1,040.8** |
| SpaceInvaders | 1,466.0 | 28,807.5 | 131.3 | 327.5 | 264.8 | **988.4** | 870.5 |
| StarGunner | 18,799.0 | 728.0 | 642.6 | **12,927.1** | 743.2 | 677.3 | 6,189.2 |
| TimePilot | 2,613.8 | 11,205.9 | 2,409.2 | 1,860.5 | 1,842.7 | 1,408.7 | **2,418.6** |
| UpNDown | 8,889.8 | 17,629.3 | 1,174.4 | 2,307.4 | 541.2 | **11,018.7** | 9,193.7 |
| VideoPinball | 87,468.5 | 269,381.0 | 5,213.2 | **12,181.1** | 6,480.1 | 4,810.8 | 7,638.2 |
| WizardOfWor | 1,904.9 | 4,749.5 | 480.3 | 1,298.0 | 882.6 | 794.7 | **1,739.4** |
| YarsRevenge | 20,520.1 | 47,218.7 | 3,089.1 | 10,056.0 | 10,047.2 | 768.4 | **21,253.5** |
| Zaxxon | 2,985.8 | 14,329.2 | 313.8 | 3,013.2 | 283.8 | 1,170.9 | **4,215.0** |

# G   Training Curves

Figure 7 presents training curves for all 46 games in RL Unplugged. Each point corresponds to the average episodic return during training binned over 1M frames. The shaded region corresponds to the 95% bootstrapped confidence interval for the mean over three runs. Note: These results will differ from Table 3 as we use a separate evaluation phase with a lower value of $\epsilon$ as is standard in the ALE.

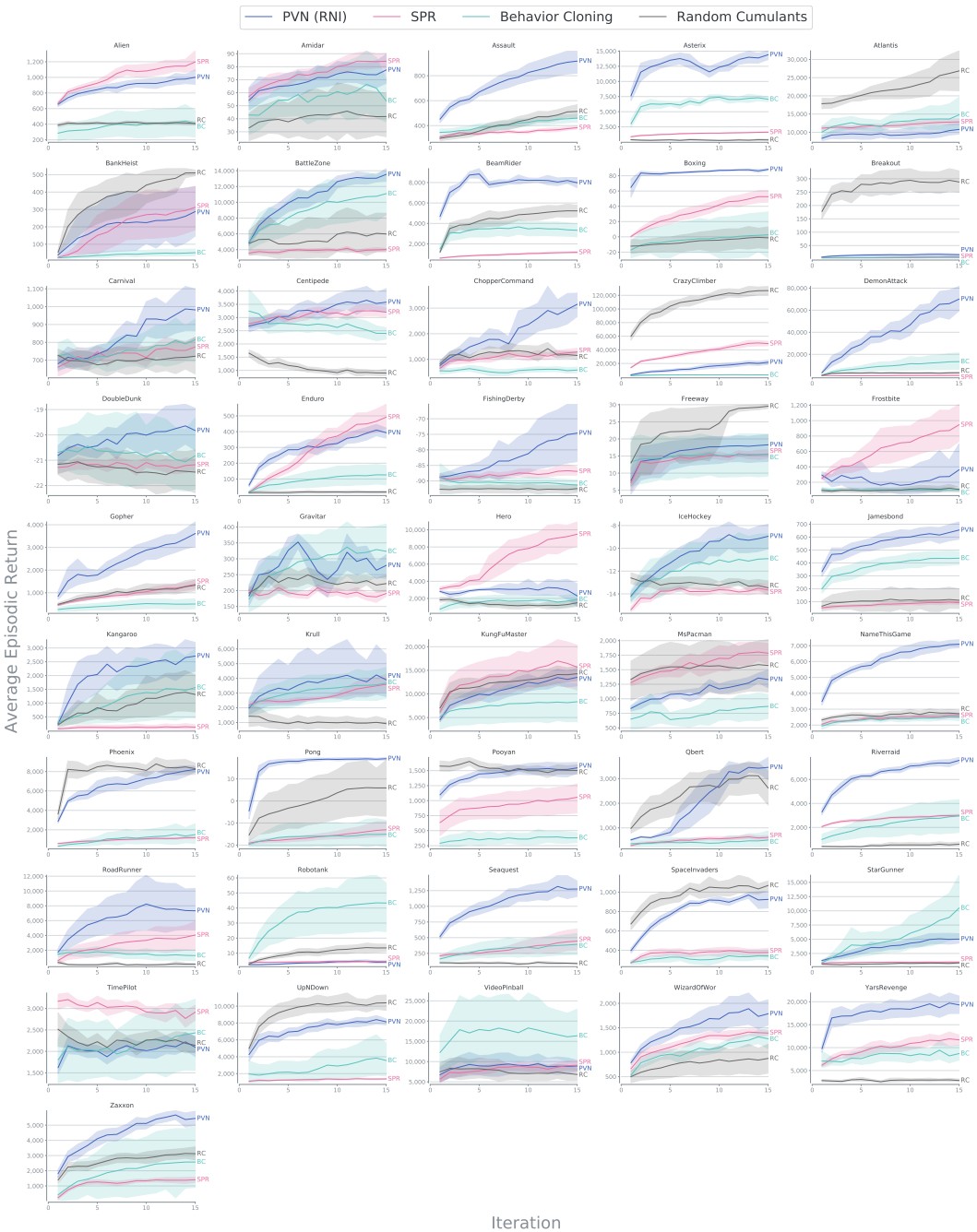

Figure 7: Training curves for: PVN (RNI), SPR, and Random Cumulants on all 46 games in RL Unplugged. The shaded region corresponds to the 95% bootstrapped confidence interval for the mean over three runs. The dashed horizontal line corresponds to the average evaluation score for Behavioral Cloning over three runs.

