# OpenReview forum: "Proto-Value Networks: Scaling Representation Learning with Auxiliary Tasks"
_NeurIPS.cc/2022/Workshop/Offline_RL — Offline RL Workshop NeurIPS 2022_

### Official Review · Reviewer_r5Gs · 2022-10-19
**The paper studies the scaling of successor measure based learned representations  with increasing number of auxiliary tasks and network size and results corroborate with prior work and expectations. Weak accept.**

**Rating:** 6
**Confidence:** 3

**Review:**

The authors mainly study how well the learned representations scale when using more auxiliary tasks and using larger networks. Particularly, they learn representations using the successor measure from Blier et al. 2021 using offline datasets.They corroborate findings from prior work showing that the best representations are learned with few auxiliary tasks and that using larger networks allows for leveraging more auxiliary tasks for improving performance. Further they visualize the learned features using MDS to show that their method produces features that better capture the state connectivity than baselines such as networks trained using the same initialization and networks trained to optimize random reward functions (Random cumulants).

Significance: Their corroboration of prior results on using appropriate number of auxiliary tasks for learning good representations is useful. Further the results on how this scales with network size and number of tasks is important.

Novelty: Their method reuses ideas from prior work but performs an analysis on how these methods scale with number of tasks and network size, which is previously missing.

Impact: Results are not surprising but the work provides a useful perspective.

Quality: The claims are well justified with experiments on Atari environments. However, using the value functions of the random policy (with respect to the randomly chosen reward) is justified suggesting that it captures the dynamics of the environment. But in that case the policy isn't necessarily random if it also explores enough to capture these dynamics.

Clarity: Sufficient. But missing an abstract and a conclusion. Would also be good to have a summary of the claims.

Reproducibility: Hyperparameters provided in the appendix.

Overall, weak accept since the results are useful but not surprising. The work could be improved with more baseline comparisons for example with the InFeR method by Lyle et al. 2022.